# Impact of Palliative Care Services on Treatment and Resource Utilization for Hepatorenal Syndrome in the United States

**DOI:** 10.3390/medicines8050021

**Published:** 2021-05-12

**Authors:** Charat Thongprayoon, Wisit Kaewput, Tananchai Petnak, Oisin A. O’Corragain, Boonphiphop Boonpheng, Tarun Bathini, Saraschandra Vallabhajosyula, Pattharawin Pattharanitima, Ploypin Lertjitbanjong, Fawad Qureshi, Wisit Cheungpasitporn

**Affiliations:** 1Department of Medicine, Mayo Clinic, Division of Nephrology and Hypertension, Rochester, MN 55905, USA; Qureshi.Fawad@mayo.edu; 2Department of Military and Community Medicine, Phramongkutklao College of Medicine, Bangkok 10400, Thailand; 3Division of Pulmonary and Pulmonary Critical Care Medicine, Faculty of Medicine, Ramathibodi Hospital, Mahidol University, Bangkok 10400, Thailand; 4Department of Medicine, Division of Pulmonary and Critical Care Medicine, Mayo Clinic, Rochester, MN 55905, USA; 5Department of Thoracic Medicine and Surgery, Temple University Hospital, Philadelphia, PA 19140, USA; 109426469@umail.ucc.ie; 6Department of Medicine, David Geffen School of Medicine, University of California, Los Angeles, CA 90095, USA; boonpipop.b@gmail.com; 7Department of Internal Medicine, University of Arizona, Tucson, AZ 85719, USA; tarunjacobb@gmail.com; 8Section of Interventional Cardiology, Department of Medicine, Division of Cardiovascular Medicine, Emory University School of Medicine, Atlanta, GA 30322, USA; saraschandra.vallabhajosyula@emory.edu; 9Department of Internal Medicine, Faculty of Medicine, Thammasat University, Pathum Thani 12120, Thailand; 10Division of Pulmonary, Critical Care and Sleep Medicine, University of Tennessee Health Science Center, Memphis, TN 38163, USA; ploypinlert@gmail.com

**Keywords:** hepatorenal syndrome, palliative care, resource utilization, outcomes, hospitalization

## Abstract

**Background:** This study aimed to determine the rates of inpatient palliative care service use and assess the impact of palliative care service use on in-hospital treatments and resource utilization in hospital admissions for hepatorenal syndrome. **Methods:** Using the National Inpatient Sample, hospital admissions with a primary diagnosis of hepatorenal syndrome were identified from 2003 through 2014. The primary outcome of interest was the temporal trend and predictors of inpatient palliative care service use. Logistic and linear regression was performed to assess the impact of inpatient palliative care service on in-hospital treatments and resource use. **Results:** Of 5571 hospital admissions for hepatorenal syndrome, palliative care services were used in 748 (13.4%) admissions. There was an increasing trend in the rate of palliative care service use, from 3.3% in 2003 to 21.1% in 2014 (*p* < 0.001). Older age, more recent year of hospitalization, acute liver failure, alcoholic cirrhosis, and hepatocellular carcinoma were predictive of increased palliative care service use, whereas race other than Caucasian, African American, and Hispanic and chronic kidney disease were predictive of decreased palliative care service use. Although hospital admission with palliative care service use had higher mortality, palliative care service was associated with lower use of invasive mechanical ventilation, blood product transfusion, paracentesis, renal replacement, vasopressor but higher DNR status. Palliative care services reduced mean length of hospital stay and hospitalization cost. **Conclusion:** Although there was a substantial increase in the use of palliative care service in hospitalizations for hepatorenal syndrome, inpatient palliative care service was still underutilized. The use of palliative care service was associated with reduced resource use.

## 1. Introduction

Hepatorenal syndrome is a severe complication of end-stage liver disease characterized by splanchnic vasodilation resulting in decreased renal blood flow [1]. Hepatorenal syndrome is classified into two types; type 1 and 2, based on the onset of disease progression. Patients with hepatorenal syndrome type 1 typically present with a rapidly progressive decline in renal function, whereas steady progression of renal impairment over weeks to months is more common in hepatorenal syndrome type 2 [1]. The prevalence of hepatorenal syndrome ranges from 13–45% in patients with end-stage liver disease developing kidney dysfunctions [2,3,4]. The prognosis of hepatorenal syndrome is associated with increased mortality, with survival ranging from months for type 2 disease, to weeks to months for type 1 [1,5]. Since hepatorenal syndrome is typically associated with long standing and greater severity of end-stage liver disease, other complications commonly coexist, such as ascites, esophageal varices, or hepatic encephalopathy [6]. These patients usually suffer from physical and psychological symptoms [7]. 

Palliative care in patients with chronic diseases other than malignancy is less often utilized in clinical practice. The purpose of palliative care is to provide support to patients, families, and caregivers by focusing of symptom management, facilitating goals of care discussion and advanced care planning, and end of life care. Hospice level of care is considered and offered for patients with a life expectancy less than 6 months [8]. Therefore, palliative care should be offered as early as end-stage diseases are identified.

Although the benefits of palliative care in end-stage liver disease have been well established, palliative care consultation remains low [8,9,10,11]. Data regarding the utilization of palliative care in patients diagnosed with hepatorenal syndrome is limited. Therefore, we conducted this study to determine the rates of inpatient palliative care service use and assess the impact of palliative care service use on in-hospital treatments and resource utilization during hospital admissions for hepatorenal syndrome.

## 2. Materials and Methods

### 2.1. Study Population

The National Inpatient Sample (NIS) database is the largest all-payer database of hospital admissions in the United States. It contains the data of more than 7 million hospital admissions per year from a 20% stratified sample of more than 4000 hospitals in the United States. The information includes patient demographics, principal diagnosis, up to 24 secondary diagnoses, and procedural codes. The database does not examine individual patients but rather examines a single inpatient admission. Because the data in the NIS are publicly available and de-identified, institutional review board approval was waived.

Using the NIS data from 2003 through 2014, a retrospective cohort of hospital admissions with a primary discharge diagnosis of hepatorenal syndrome based on International Classification of Diseases, Ninth Edition (ICD-9) diagnosis code of 572.4, was built. 

### 2.2. Data Collection

The primary variable of interest was palliative care service. Palliative care service was identified using ICD-9 of V66.7. This code was added to discharge data when there was a billing for palliative care options such as end of life, hospice, and comfort care in medical records. However, it did not include pain and symptom management. The ICD-9 code-based identification of palliative care service was validated and used in previous studies and showed moderate sensitivity and high specificity of more than 90% [12,13,14]. 

The following clinical characteristics were abstracted from the database (Appendix A): age, sex, race, year of hospital admissions, smoking history, alcohol use, obesity, diabetes mellitus, hypertension, hyperlipidemia, coronary artery disease, atrial flutter/fibrillation, congestive heart failure, chronic kidney disease, acute liver failure, hepatitis B infection, hepatitis C infection, alcoholic cirrhosis, non-alcoholic steatohepatitis, and hepatocellular carcinoma. In-hospital acute events included gastrointestinal bleeding, septic shock, bloodstream infection, cardiac arrest, and organ failure. In-hospital treatment included invasive mechanical ventilation, non-invasive ventilation support, blood product transfusion, enteral nutrition, paracentesis, renal replacement therapy, vasopressor use, and do-not-resuscitate (DNR) orders. Resource use included length of hospital stay and hospitalization cost. Since this study used a dataset spanning 12 calendar years, hospitalization cost was adjusted for inflation using the consumer price index and values were converted to their 2014 US dollar equivalent. 

### 2.3. Statistical Analysis

Survey procedures using discharge weights provided by the NIS database were performed to generate national estimates. The temporal trend in the rate of palliative care service in hospital admissions for hepatorenal syndrome in the United States from 2003 to 2014 was graphically displayed by estimating proportion with their standard errors and was tested using an ordinary least square regression model with the year of hospital admissions as the predictor. Student’s *t*-test and Chi-squared test were used to compare continuous and categorical variables between hospital admissions with and without palliative care service, respectively. Multivariable logistic regression with backward stepwise selection was performed to evaluate predictors of inpatient palliative care service. The impact of palliative care service on in-hospital treatments was examined using logistic regression, whereas the impact on resource use was examined using linear regression. The analyses were adjusted for variables that significantly differed between hospital admissions with and without palliative care service in univariable analysis. Two-tailed *p*-value < 0.05 was considered statistically significant. SPSS statistical software (version 22.0, IBM Corporation, Armonk, NY, USA) was used for all analyses.

## 3. Results

### 3.1. The Rate of and Trend in Palliative Care Service Use in Hospital Admissions for Hepatorenal Syndrome

There were 5571 hospital admission for hepatorenal syndrome in the United States from 2003 to 2014. Of these admissions, 748 (13.4%) had palliative care service use. Figure 1 showed the rate of palliative care service use during hospital admission for hepatorenal syndrome in the United States from 2003 to 2014. There was a steady increase in the rate of palliative care service use from 3.3% in 2003 to 21.1% in 2014 (*p*-trend < 0.001).

### 3.2. The Predictors of Inpatient Palliative Care Service Use in Hospital Admissions for Hepatorenal Syndrome

Table 1 compares clinical characteristics, treatments, and resource use between hospital admissions for hepatorenal syndrome with and without palliative care service use. In multivariable analysis, age ≥ 60 years (OR 1.47; 95% CI 1.15–1.88 for age 60–69 years, and 1.88; 95% CI 1.44–2.44 for age ≥ 70 years, compared to age < 50 years), more recent year of hospital admissions (OR 2.95; 95% CI 2.19–3.98 for year 2007–2010, and 4.59; 95% CI 3.45–6.10 for year 2011–2014, compared to year 2003–2006), acute liver failure (OR 1.24; 95% CI 1.05–1.46), alcoholic cirrhosis (OR 1.26; 95% CI 1.07–1.49), and hepatocellular carcinoma (OR 1.73; 95% CI 1.31–2.28) were predictive of increased palliative care service use, whereas race other than Caucasian, African American, and Hispanic (OR 0.80; 95% CI 0.63–0.99) and chronic kidney disease (OR 0.65; 95% CI 0.54–0.78) were predictive of decreased palliative care service use (Table 2). In-hospital acute events or organ failure were not predictive of palliative care service use.

### 3.3. The Impact of Inpatient Palliative Care Service Use on Treatments and Resource Use in Hospital Admissions for Hepatorenal Syndrome

Table 3 shows in-hospital treatments and resource use between hospital admissions with and without palliative care service use. Hospital admissions with palliative care service use had significantly lower use of invasive mechanical ventilation (7.9% vs. 10.7%; *p* = 0.02), blood product transfusion (23.4% vs. 33.0%; *p* < 0.001), paracentesis (32.8% vs. 45.2%; *p* < 0.001), renal replacement therapy (12.4% vs. 21.3%; *p* < 0.001), vasopressor use (34.4% vs. 46.3%; *p* < 0.001) but higher DNR status (24.5% vs. 4.1%; *p* < 0.001). The length of hospital stay and hospitalization cost was significantly lower in hospital admissions with palliative care service use.

Despite adjustment for the difference in clinical characteristics, palliative care service use was associated with decreased use of invasive mechanical ventilation (OR 0.72; 95% CI 0.54–0.96), blood product transfusion (OR 0.59; 95% CI 0.49–0.71), paracentesis (OR 0.50; 95% CI 0.42–0.59), renal replacement therapy (OR 0.52; 95% CI 0.41–0.66), vasopressor use (OR 0.51; 95% CI 0.43–0.60) but was associated with increased use of DNR order (OR 5.95; 95% CI 4.68–7.56). The use of palliative care service was associated with reduced mean length of hospital stay by 2.1 days (*p* < 0.001) and reduced mean hospitalization cost by $26,268 (; *p* < 0.001) (Table 4). 

Hospital admissions with palliative care service use had higher mortality rates (54.1% vs. 30.1%). The reduction in resource use was consistently noted in subgroup analysis based on vital status at hospital discharge. However, the reduction in length of hospital stay associated with palliative care service use was more prominent in patients who were discharged from hospital alive (mean difference −2.1 in alive vs. −1.0 days in dead patients; *p*-interaction < 0.001). The reduction in hospitalization cost associated with palliative care service use was more prominent in patients who died in hospital (mean difference −$24,413 in dead vs. −$22,761 in alive patients; *p*-interaction < 0.001).

## 4. Discussion

In this large hospitalized database cohort, we demonstrated a trend of increased palliative care service use in hospitalized patients with hepatorenal syndrome during the study period. The overall rate of palliative care service use was 13.4%. Variables associated with increased palliative care service use included more recent hospitalization year, older age, hepatocellular carcinoma, alcoholic cirrhosis, and acute liver failure. In contrast, races other than Caucasian, African American, and Hispanic and chronic kidney disease were associated with decreased palliative care service use. Even though higher mortality was observed, palliative care service use was associated with less invasive procedures, decreased length of hospital stays, and decreased hospitalization cost.

The benefits of palliative care in non-cancer-related chronic diseases have been increasingly recognized over time. The reported rates of palliative care consultation in end-stage chronic liver disease are variable. Previous studies demonstrated palliative care consultation rates ranging from 29.1% to 48.6% for inpatient admissions with end-stage liver disease, though only 19% received early palliative care, defined as 30 days or more prior to death [9,10]. Although palliative care service use in our study appears lower than previous reports, the rate of palliative care service use in our study consistently increased across the study period, increasing from 3.3% in 2003 to 21.1% in 2014. A potential reason to explain lower rates of referral is that our study included hospitalized patients from 2003 when palliative care was less well developed for patients with non-cancer disease.

Our study identified several variables associated with increased or decreased palliative care service use in hospitalized patients with hepatorenal syndrome. Similar to our study, a previous single-center study from the United States also identified that older age and hepatocellular carcinoma predicted increased rates of palliative care consultation in end-stage liver disease [11]. These patients often do not meet criteria for consideration of liver transplantation and, thus, palliative care is appropriate [15]. Although an association between alcoholic liver disease and decreased palliative care consultation was previously reported [9], our study revealed increased palliative care consultation in this subgroup. Furthermore, our study demonstrated that acute liver failure is also associated with increased palliative care service use. Although our study demonstrated that several acute conditions did not predict palliative care consultation, high mortality of up to 50% associated with acute liver failure supports the important role of palliative care [16]. 

In contrast, races other than Caucasian, African American, and Hispanic were associated with decreased palliative care service use. Cultural and religious factors are important considerations influencing decision making to accept palliative care [17]. Discussion on forthcoming death is considered inappropriate and a sensitive topic in some cultures. Individualized and tailored communication regarding the role of palliative care sensitive to a patient’s religious and cultural background may encourage palliative care among patients. Chronic kidney disease also appeared to be associated with decreased palliative care service use. Chronic kidney disease is more likely to be associated with hepatorenal syndrome type 2, which has a more favorable outcome than type 1 disease [1]. Therefore, palliative care might be less of a concern among these patients. Furthermore, the rates of combined liver–kidney transplantation have increased over time, accounting for 10% of all liver transplants in the United States [18]. Patients with concurrent chronic kidney disease and end-stage liver disease may be considered for combined transplantation, perhaps decreasing rates of palliative care use if actively listed for transplant or pursuing transplant evaluation. 

The potential benefits of palliative care in end-stage liver disease include improvements in the quality of life, lower readmission rates, decreased procedure burden, and lower costs of hospitalization [10,19,20]. Our study demonstrated higher mortality in patients receiving palliative care services, though with reduced invasive procedures performed, higher rates of DNR status and lower hospitalization costs. As palliative care consultation is more likely in patients with more advanced or terminal disease, it is likely those who received palliative care represented a higher risk subgroup. In patients with hepatorenal syndrome with an overall high expected mortality rate, palliative care service use is of critical importance for patients and their families to establish patient-centered goals of care and limit intensive care level interventions, such as invasive mechanical ventilation and vasopressor use, as seen in our study.

The NIS database is dependent on discharge diagnostic codes and, therefore, some limitations should be noted in this study. Firstly, the ICD code for hepatorenal syndrome might not be accurate since the diagnostic criteria for this condition are based on the exclusion of alternative causes of renal impairment. Furthermore, subgroup analyses stratified by severity of disease or subtypes of hepatorenal syndrome was not possible as the NIS database does not provide laboratory tests and does not code hepatorenal syndrome subtypes. Secondly, symptoms (e.g., pain, anxiety) of patients with hepatorenal syndrome would give more information about the indication of inpatient palliative service because the indication for palliative care service is generally based on symptoms, not diagnosis itself. Unfortunately, the NIS database did not contain information on symptoms or severity of symptoms given its nature of being a billing database and, thus, it limited the value of our findings. Thirdly, this study only highlighted the increased rate of palliative care service and the reduced length of stay and hospitalization cost with palliative care service in hospital admissions for hepatorenal syndrome, but we did not evaluate impact of palliative care service on key elements of quality of life measures or symptom relief, which are fundamental goals of palliative care. In addition, we did not assess for long-term outcomes among patients discharged from the hospital, such as readmission rates, invasive procedure burden after discharge, or long-term mortality. Future studies would benefit greatly from adding a quality of life component or by assessing long-term outcomes. 

## 5. Conclusions

Although palliative care service use remains underutilized in patients hospitalized for hepatorenal syndrome, the rates of palliative care service use appear to have increased over time. Palliative care service use was higher in older patients and those with concurrent hepatocellular carcinoma, alcoholic cirrhosis, and liver failure. Palliative care service use was associated with decreased invasive procedure burden, length of hospital stay, and hospitalization costs. 

## Figures and Tables

**Figure 1 medicines-08-00021-f001:**
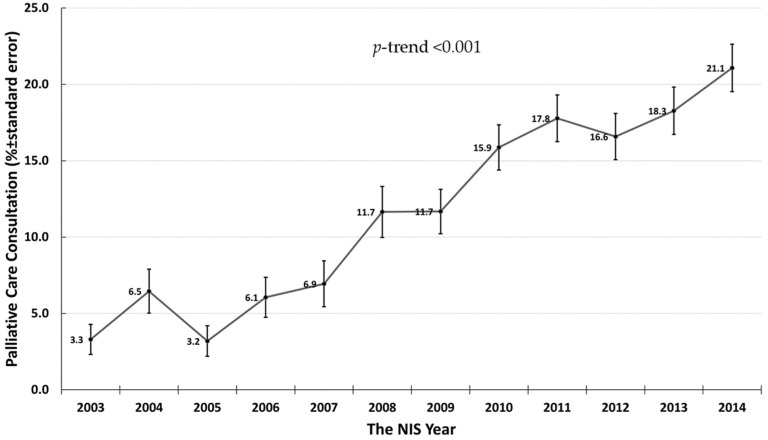
The rate of palliative care service use in hospital admission for hepatorenal syndrome in United States from 2003 to 2014.

**Table 1 medicines-08-00021-t001:** Clinical characteristics, in-hospital events, and organ dysfunction between hospital admissions for hepatorenal syndrome with and without palliative care service use.

Clinical Characteristics	Total	Palliative Care	No Palliative Care	*p*-Value
N	5571	748	4823	
Age (years), mean ± SD	58.8 ± 12.6	60.4 ± 12.4	58.5 ± 12.6	<0.001
<50	1216 (21.8%)	129 (17.2%)	1087 (22.5%)	0.002
50–59	1852 (33.2%)	241 (32.2%)	1611 (33.4%)	
60–69	1403 (25.2%)	208 (27.8%)	1195 (24.8%)	
≥70	1100 (19.7%)	170 (22.7%)	930 (19.3%)	
Male	3537 (63.5%)	493 (65.9%)	3044 (63.1%)	0.14
Race				
Caucasian	3422 (61.4%)	504 (67.4%)	2918 (60.5%)	0.001
African American	473 (8.5%)	59 (7.9%)	414 (8.6%)	
Hispanic	584 (10.5%)	77 (10.3%)	507 (10.5%)	
Other	1092 (19.6%)	108 (14.4%)	984 (20.4%)	
Year of hospitalization				
2003–2006	1270 (22.8%)	60 (8.0%)	1210 (25.1%)	<0.001
2007–2010	1756 (31.5%)	217 (29.0%)	1539 (31.9%)	
2011–2014	2545 (45.7%)	471 (63.0%)	2074 (43.0%)	
Smoking	446 (8.0%)	65 (8.7%)	381 (7.9%)	0.46
Alcohol drinking	1796 (32.2%)	263 (35.2%)	1533 (31.8%)	0.07
Obesity	340 (6.1%)	46 (6.1%)	294 (6.1%)	0.95
Diabetes mellitus	1184 (21.3%)	155 (20.7%)	1029 (21.3%)	0.70
Hypertension	1739 (31.2%)	217 (29.0%)	1522 (31.6%)	0.16
Hyperlipidemia	352 (6.3%)	43 (5.7%)	309 (6.4%)	0.49
Coronary artery disease	467 (8.4%)	56 (7.5%)	411 (8.5%)	0.34
Atrial flutter/fibrillation	400 (7.2%)	47 (6.3%)	353 (7.3%)	0.31
Congestive heart failure	680 (12.2%)	73 (9.8%)	607 (12.6%)	0.03
Chronic kidney disease	1620 (29.1%)	192 (25.7%)	1428 (29.6%)	0.03
Acute liver failure	2247 (40.3%)	345 (46.1%)	1902 (39.4%)	0.001
Hepatitis B infection	149 (2.7%)	13 (1.7%)	136 (2.8%)	0.09
Hepatitis C infection	1208 (21.7%)	148 (19.8%)	1060 (22.0%)	0.18
Alcoholic cirrhosis	2081 (37.4%)	312 (41.7%)	1769 (36.7%)	0.008
Non-alcoholic steatohepatitis	1069 (19.2%)	133 (17.8%)	936 (19.4%)	0.29
Hepatocellular carcinoma	343 (6.2%)	78 (10.4%)	265 (5.5%)	<0.001
Hospital events				
Gastrointestinal bleeding	689 (12.4%)	83 (11.1%)	606 (12.6%)	0.26
Septic shock	198 (3.6%)	31 (4.1%)	167 (3.5%)	0.35
Bloodstream infections	641 (11.5%)	70 (9.4%)	571 (11.8%)	0.04
Cardiac arrest	187 (3.4%)	22 (2.9%)	165 (3.4%)	0.50
In hospital mortality	1857 (33.3%)	405 (54.1%)	1452 (30.1%)	<0.001
Organ dysfunction				
Respiratory failure	920 (16.5%)	119 (15.9%)	801 (16.6%)	0.63
Circulatory failure	1070 (19.2%)	157 (21.0%)	913 (18.9%)	0.18
Metabolic failure	1371 (24.6%)	190 (25.4%)	1181 (24.5%)	0.59
Neurological failure	447 (8.0%)	70 (9.4%)	377 (7.8%)	0.15
Hematological failure	1987 (35.7%)	262 (35.0%)	1725 (35.8%)	0.69

Continuous data are presented as mean ± SD or median (IQR); categorical data are presented as count (%).

**Table 2 medicines-08-00021-t002:** Predictors of inpatient palliative service use in hospital admissions for hepatorenal syndrome.

Variables	Univariable Analysis		Multivariable Analysis	
	OR (95% CI)	*p*-Value	Adjusted OR (95% CI)	*p*-Value
Clinical characteristics				
Age (years)				
<50	1 (reference)		1 (reference)	
50–59	1.26 (1.01–1.58)	0.04	1.22 (0.97–1.54)	0.10
60–69	1.47 (1.16–1.85)	0.001	1.47 (1.15–1.88)	0.002
≥70	1.54 (1.21–1.97)	0.001	1.88 (1.44–2.44)	<0.001
Male	1.13 (0.96–1.33)	0.14		
Race				
Caucasian	1 (reference)		1 (reference)	
African American	0.83 (0.62–1.10)	0.19	0.89 (0.66–1.19)	0.43
Hispanic	0.88 (0.68–1.14)	0.33	0.92 (0.70–1.19)	0.52
Other	0.64 (0.51–0.79)	<0.001	0.80 (0.63–0.99)	0.04
Year				
2003–2006	1 (reference)		1 (reference)	
2007–2010	2.84 (2.12–3.82)	<0.001	2.95 (2.19–3.98)	<0.001
2011–2014	4.58 (3.47–6.05)	<0.001	4.59 (3.45–6.10)	<0.001
Smoking	1.11 (0.84–1.46)	0.46		
Alcohol drinking	1.16 (0.99–1.37)	0.07		
Obesity	1.01 (0.73–1.39)	0.95		
Diabetes mellitus	0.96 (0.80–1.17)	0.70		
Hypertension	0.88 (0.75–1.05)	0.16		
Dyslipidemia	0.89 (0.64–1.24)	0.49		
Coronary artery disease	0.87 (0.65–1.16)	0.34		
Atrial flutter/fibrillation	0.85 (0.62–1.16)	0.31		
Congestive heart failure	0.75 (0.58–0.97)	0.03		
Chronic kidney disease	0.82 (0.69–0.98)	0.03	0.65 (0.54–0.78)	<0.001
Acute liver failure	1.32 (1.13–1.54)	0.001	1.24 (1.05–1.46)	0.01
Hepatitis B infection	0.61 (0.34–1.08)	0.09		
Hepatitis C infection	0.88 (0.72–1.06)	0.18		
Alcoholic cirrhosis	1.24 (1.06–1.45)	0.008	1.26 (1.07–1.49)	0.007
Non-alcoholic steatohepatitis	0.90 (0.74–1.10)	0.29		
Hepatocellular carcinoma	2.00 (1.54–2.61)	<0.001	1.73 (1.31–2.28)	<0.001
Hospital events and organ dysfunction				
Gastrointestinal bleeding	0.87 (0.68–1.11)	0.26		
Septic shock	1.21 (0.82–1.78)	0.35		
Bloodstream infections	0.77 (0.59–0.99)	0.04		
Cardiac arrest	0.86 (0.55–1.34)	0.50		
Respiratory failure	0.95 (0.77–1.17)	0.63		
Circulatory failure	1.14 (0.94–1.38)	0.18		
Metabolic failure	1.05 (0.88–1.25)	0.59		
Neurological failure	1.22 (0.93–1.59)	0.15		
Hematological failure	0.97 (0.82–1.14)	0.69		

**Table 3 medicines-08-00021-t003:** In-hospital treatments and resource use between hospital admissions for hepatorenal syndrome with and without palliative care service use.

Treatments	Total	Palliative Care	No Palliative Care	*p*-Value
Invasive mechanical ventilation	575 (10.3)	59 (7.9)	516 (10.7)	0.02
Non-invasive ventilation	94 (1.7)	13 (1.7)	81 (1.7)	0.91
Blood product transfusion	1765 (31.7)	175 (23.4)	1590 (33.0)	<0.001
Enteral nutrition	97 (1.7)	11 (1.5)	86 (1.8)	0.54
Paracentesis	2423 (43.5)	245 (32.8)	2178 (45.2)	<0.001
Renal replacement therapy	1122 (20.1)	93 (12.4)	1029 (21.3)	<0.001
Vasopressor	2491 (44.7)	257 (34.4)	2234 (46.3)	<0.001
Do-not-resuscitate status	379 (6.8)	183 (24.5)	196 (4.1)	<0.001
Resource utilization				
Length of hospital stay (days), mean ± SD	8.7 ± 10.7	6.9 ± 8.4	9.0 ± 11.0	<0.001
Hospital cost ($), mean ± SD	69,765 ± 130,188	52,929 ± 82,404	72,340 ± 135,848	<0.001

**Table 4 medicines-08-00021-t004:** The impact of inpatient palliative care service use on treatments and resource use in hospital admissions for hepatorenal syndrome.

Treatments	Adjusted Odds Ratio * (95% CI)	*p*-Value
Invasive mechanical ventilation	0.72 (0.54–0.96)	0.02
Non-invasive ventilation	0.97 (0.53–1.77)	0.91
Blood product transfusion	0.59 (0.49–0.71)	<0.001
Enteral nutrition	0.73 (0.38–1.38)	0.33
Paracentesis	0.50 (0.42–0.59)	<0.001
Renal replacement therapy	0.52 (0.41–0.66)	<0.001
Vasopressor	0.51 (0.43–0.60)	<0.001
Do-not-resuscitate status	5.95 (4.68–7.56)	<0.001
Resource use	Adjusted coefficient * (95% CI)	
Length of stay (days)	−2.1 (−2.9 to −1.3)	<0.001
Hospital cost ($)	−26,268 (−36,429 to −16,107)	<0.001

* Adjusted for age, race, year of hospitalization, congestive heart failure, chronic kidney disease, acute liver failure, alcoholic cirrhosis, and hepatocellular carcinoma.

## Data Availability

Restrictions apply to the availability of these data. Data was obtained from the Healthcare Cost and Utilization Project (HCUP) under the sponsorship of the Agency for Healthcare Research and Quality (AHRQ) and are available https://www.distributor.hcup-us.ahrq.gov/ with the permission of HCUP/AHRQ.

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
