# Peer review of "Impact of Palliative Care Services on Treatment and Resource Utilization for Hepatorenal Syndrome in the United States"

_medicines, 2021, doi:10.3390/medicines8050021_

Round 1

Reviewer 1 Report

Dear authors, 

the submitted manuscript displays the use of inpatient palliative services in the US between 2003 and 2014 among patients with hepatorenal syndrome. It shows interesting results that are in general not surprising and integrate well in the known body of evidence about the utilisation of palliative care services in hospitals.

The methodolody is appropiate, the results are displayed in a structured way and the discussion aligns the findings with the literature. Of course, some aspects could be discussed in more depths (e.g. cultural aspects, late vs. early integration of palliative care services etc.) but that might be beyond the aim of this paper. 

My main critique concerns the correlation between diagnosis and use of palliative services. I am not sure, if this gives valuable information. Indication for Palliative Care is based on symptoms (or severity of symptoms) and not diagnosis. Instead of using the ICD-9 "diagnosis codes", you could have used "symptoms codes".  I recognize that you wrote, NIS records only the "discharge diagnosis". Does this means, symptoms are not coded in the NIS database? If you have access to these data, it would be a good asset for this study to include this or exchange "Diagnosis" with "Symptoms", respectively. Symptoms (pain, anxiety, etc.) would give more information about the indication of inpatient palliative services.

Otherwise, this is a very sound piece of research and I have no further objections for publication.

Author Response

Response to Reviewer# 1

the submitted manuscript displays the use of inpatient palliative services in the US between 2003 and 2014 among patients with hepatorenal syndrome. It shows interesting results that are in general not surprising and integrate well in the known body of evidence about the utilisation of palliative care services in hospitals.

The methodolody is appropiate, the results are displayed in a structured way and the discussion aligns the findings with the literature. Of course, some aspects could be discussed in more depths (e.g. cultural aspects, late vs. early integration of palliative care services etc.) but that might be beyond the aim of this paper.

Response: We thank you for reviewing our manuscript and critical evaluation.

Comment #1

My main critique concerns the correlation between diagnosis and use of palliative services. I am not sure, if this gives valuable information. Indication for Palliative Care is based on symptoms (or severity of symptoms) and not diagnosis. Instead of using the ICD-9 "diagnosis codes", you could have used "symptoms codes".  I recognize that you wrote, NIS records only the "discharge diagnosis". Does this means, symptoms are not coded in the NIS database? If you have access to these data, it would be a good asset for this study to include this or exchange "Diagnosis" with "Symptoms", respectively. Symptoms (pain, anxiety, etc.) would give more information about the indication of inpatient palliative services.

Response #1: We agreed with your suggestion that the data on patients’ symptoms would be very informative when we assessed the indication for palliative care service. However, symptom codes are not available in NIS database. The following statements have been added in the limitation section.

“symptoms (e.g., pain, anxiety) of patients with hepatorenal syndrome would give more information about the indication of inpatient palliative service because the indication for palliative care service is generally based on symptoms, not diagnosis itself. Unfortunately, NIS database did not contain information on symptoms or severity of symptoms given its nature of billing database, and thus, it limited the value of our findings.”

Comment #2

Otherwise, this is a very sound piece of research and I have no further objections for publication.

Response: We greatly appreciated the reviewer’s time and comments to improve our manuscript. The manuscript has been improved considerably by the suggested revisions.

We greatly appreciated the reviewer’s and editor’s time and comments to improve our manuscript. The manuscript has been improved considerably by the suggested revisions.

Reviewer 2 Report

In the submitted manuscript “Impact of Palliative Care Services on Treatment and Resource Utilization for Hepatorenal Syndrome in the United States“, the authors examine the rates of inpatient palliative care service use in hospital admissions for hepatorenal syndrome.

The article is interesting, however it does have several limitations, most of which result from the study design itself (limited data extracted from a single NIS database) and are mentioned by the authors at the end of the discussion section.

It must be stressed, that this study only highlights the increased rate of palliative care service and reduced LoS and cost without evaluating its impact on key elements of quality-of-life measures or symptom relief, which are fundamental goals of palliative care as written in the discussion (line 261). Thus, it would benefit greatly by adding a QoL component or by assessing long-term outcomes. This would require significantly more effort.

There is a double point in lines 213, 250 - error?

In conclusion, the paper is well-written, clearly describes research design, materials and methods, as well as the results.

Author Response

Response to reviewer #2

In the submitted manuscript “Impact of Palliative Care Services on Treatment and Resource Utilization for Hepatorenal Syndrome in the United States“, the authors examine the rates of inpatient palliative care service use in hospital admissions for hepatorenal syndrome.

The article is interesting, however it does have several limitations, most of which result from the study design itself (limited data extracted from a single NIS database) and are mentioned by the authors at the end of the discussion section.

Response: We thank you for reviewing our manuscript and critical evaluation.

Comment #1

It must be stressed, that this study only highlights the increased rate of palliative care service and reduced LoS and cost without evaluating its impact on key elements of quality-of-life measures or symptom relief, which are fundamental goals of palliative care as written in the discussion (line 261). Thus, it would benefit greatly by adding a QoL component or by assessing long-term outcomes. This would require significantly more effort.

Response: The following statements have been added in the limitation section as suggested.

“This study only highlighted the increased rate of palliative care service and the reduced length of stay and hospitalization cost with palliative care service in hospital admissions for hepatorenal syndrome but we did not evaluate impact of palliative care service on key elements of quality of life measures or symptom relief, which are fundamental goals of palliative care. In addition, we did not assess for long-term outcomes among patients discharged from the hospital, such as readmission rates, invasive procedure burden after discharge, or long-term mortality Future study would greatly benefit by adding quality of life component or by assessing long-term outcomes.”

Comment #2

There is a double point in lines 213, 250 - error?

            Response: We have corrected this error as suggested.

Comment #2

In conclusion, the paper is well-written, clearly describes research design, materials and methods, as well as the results.

Response: We greatly appreciated the reviewer’s time and comments to improve our manuscript. The manuscript has been improved considerably by the suggested revisions.

We greatly appreciated the reviewer’s and editor’s time and comments to improve our manuscript. The manuscript has been improved considerably by the suggested revisions.

Round 2

Reviewer 1 Report

Thank you for addressing the critique. I have no further comments.